# Indigenous People’s Use of a Primary Urgent Care Centre at a GP-Led Primary Healthcare Service in Regional Queensland in 2020–2021

**DOI:** 10.3390/ijerph22070998

**Published:** 2025-06-25

**Authors:** Shauna Fjaagesund, Wenwen Zang, Raymond Gadd, Jayley Hart, Piotr Swierkowski, Andrew Ladhams, Christopher Hicks, Sylvia Andrew-Starkey, Evan Jones, Alexandru Coman, Gavin Beccaria, Florin Oprescu, Xiang-Yu Hou

**Affiliations:** 1School of Health, University of the Sunshine Coat, Sippy Downs, QLD 4556, Australia; sfjaages@usc.edu.au (S.F.); evan@ewljones.com (E.J.); foprescu@usc.edu.au (F.O.); 2General Practice Unit, Faculty of Medicine, University of Queensland, Brisbane, QLD 4072, Australia; 3Poche Centre for Indigenous Health, University of Queensland, Brisbane, QLD 4066, Australia; wenwenzang@hotmail.com; 4Health Hub Doctors Morayfield, Morayfield, QLD 4506, Australia; raygadd@me.com (R.G.); nm@hhdm.com.au (J.H.); peter_swierkowski@hotmail.com (P.S.); hicks.m.chris@gmail.com (C.H.); 5Morayfield Minor Accident and Illness Centre, Morayfield, QLD 4506, Australia; sylviaandrewstarkey@gmail.com; 6Department of Infectious Diseases and Epidemiology, University of Medicine and Pharmacy ‘Iuliu Hatieganu’, 400347 Cluj-Napoca, Romania; acoman@gmail.com; 7School of Psychology and Wellbeing, University of Southern Queensland, Toowoomba, QLD 4350, Australia; gavin.beccaria@unisq.edu.au; 8Centre for International Development, Social Entrepreneurship and Leadership, University of the Sunshine Coast, Sippy Downs, QLD 4556, Australia; 9Centre for Health Research, University of Southern Queensland, Toowoomba, QLD 4350, Australia; 10Susan Wakil School of Nursing and Midwifery, Broken Hill University Department of Rural Health, Faculty of Medicine and Health, University of Sydney, Sydney, NSW 2880, Australia

**Keywords:** Aboriginal and Torres Strait Islanders, Indigenous people, general practice, primary healthcare, primary urgent care, community urgent care

## Abstract

To explore Indigenous patients’ use of a primary urgent care centre (PUCC) at a co-located general medical practitioner (GP)-led primary healthcare service (GP service) in regional Queensland, Australia, secondary data analysis was conducted using the 65,420 deidentified PUCC patients from 1 July 2020 to 30 June 2021, including Indigenous status. A Mann–Whitney U test and Chi-Square test were used to analyse patients’ arrival times, reasons to attend PUCC, and frequency of attendance. The proportion of Indigenous patients from the communities attending the PUCC was 9.8% while the proportion of Indigenous people in the general population was only 3.8%. Indigenous patients were more likely to be new patients to the GP service (13.6% never visited the GP service prior to PUCC) compared to non-Indigenous (9.6%) patients. The peak hours of attendance for Indigenous people were 11 a.m.–12 p.m. and 2 p.m.–3 p.m. while it was 10 a.m.–12 p.m. for non-Indigenous patients. The most common reason for attending PUCC for both patient groups was superficial injuries. The second most common reason was digestive issues for Indigenous patients and musculoskeletal issues for non-Indigenous patients. These findings provide insights for enhancing future PUCC models to better meet the community needs, especially the underserved Indigenous population in regional areas.

## 1. Introduction

The health gap between Aboriginal and Torres Strait Islander (hereafter respected referred to as Indigenous) and non-Indigenous Australians remains a significant and concerning issue [1]. Indigenous people face higher risk of developing chronic and preventable conditions [2], leading to an increased disease burden of 2.3 times the rate of non-Indigenous Australians. In North Brisbane in Queensland, Australia, Indigenous people experienced significantly poorer health outcomes, with 1.9 times higher burden of disease and injury, compared to their non-Indigenous counterparts [2,3]. Furthermore, over 70% of the recognised disease burden among Indigenous populations in North Brisbane was linked to major contributing causes, including mental and behavioural disorders, cardiovascular diseases, diabetes, chronic respiratory diseases, cancers, and neurological/sensory organ disorders [2,3].

Barriers to primary healthcare access are a contributing factor to the observed health gap in Indigenous populations [4]. The reported barriers to healthcare include reduced income levels to attend or pay for services and lower levels of educational attainment which can negatively affect health literacy needed to effectively navigate primary healthcare systems [5]. Additionally, past experiences of discrimination or perceptions of culturally unsafe environments may discourage Indigenous patients from seeking timely healthcare services [5,6,7,8,9].

Reduced engagement with primary healthcare services [1] is known to culminate into a higher frequency of emergency department (ED) service use by Indigenous populations [2,10]. In some cases, it has made hospital EDs the main access point of care for some individuals [4], including Indigenous patients [5,8]. The reasons behind this include patients’ low socioeconomic status (for the free ED care) [11], undiagnosed and unmanaged chronic conditions especially in Indigenous patients [8,12], unmet needs in terms of holistic care including physical, social, spiritual, and emotional wellbeing [7,13], and the patients’ choice to attend ED for non-life-threatening care [14]. Unfortunately, the research in ED care demonstrated that Indigenous people are more likely to leave without being seen or discharged against medical advice, in cases with low-acuity urgent and non-life-threatening health problems [14]. Other factors included ED overcrowding and long wait times which affected leaving without care for all patient types.

To address the issue of hospital ED overcrowding and long wait times by individuals who could not afford private urgent care fees, the Commonwealth of Australia [15,16] introduced public funding for GP services to deliver primary urgent care centres (PUCCs) at no cost to patients. One of these funded centres was located within a large-scale 15,000 sqm integrated primary healthcare facility [17], servicing a usual resident population of 476,340 in Moreton Bay area, North Brisbane, Queensland, Australia [18]. This PUCC provided a timely unique opportunity in Australia to better understand its use by the communities due to its high patient load and previous well-established general medical practitioner (GP)-led model of timely primary healthcare for minor injuries and illnesses [17].

The aim of this study was to use the secondary PUCC data from 2020 to 2021 to explore the Indigenous patients’ use of this unique PUCC in a GP-led primary healthcare service (GP service). The findings of the study could provide insights for health managers, clinicians, and researchers regarding policy and practice to enhance the primary acute care system and reduce the use of hospital emergency departments while improving healthcare access for Indigenous and other underserved populations.

## 2. Materials and Methods

The study was undertaken at an accredited PUCC [19] staffed by general medical practitioners (GPs), nurse practitioners, and practice nurses with previous training and experience in emergency medicine, rural and family medicine, and urgent care [19,20], using a retrospective cross-sectional study design.

In the PUCC, patients access free, walk-in urgent care services and receive a discharge diagnosis and treatment for minor accidents and illnesses, with follow-up referral for further care [20]. Agreements are in place between the PUCC and on-site pathology and radiology providers to enable immediate patient access to diagnostics with fast turnaround reporting to enable immediate diagnosis. In cases where patients were at serious health risk, ambulance transport to a major secondary hospital was arranged [19].

The PUCC used the database of the Best Practice electronic patient record platform to store medical data, in which patient information was entered by healthcare practitioners [21], and where patients could choose to consent for their information to be used for health research. The deidentified data of all attending patients with consent from 1 July 2020 to 30 June 2021 was extracted.

Demographic variables included age (years), residential postcode, sex (male or female), Indigenous status, and type of patient (new or existing patient of the GP service). The patients’ PUCC use variables included time and date and reasons of attendance. The 2021 Index of Relative Socioeconomic Advantage and Disadvantage (IRSAD) measure was used to measure the average household income and occupation skill level of patients living in their suburbs using their post codes [22]. The reasons for PUCC attendance were categorised in accordance with the Australian Emergency Care Classification (AECC) system, which was based on the ICD-10-AM/ACHI/ACS Eleventh Edition.

The continuous variables did not follow a normal distribution, and a non-parametric analysis approach was adopted. For comparisons of two groups (Indigenous vs. Non-Indigenous), a Mann–Whitney U test and Chi-Square test were used in the data analysis. The data analysis was undertaken using IBM SPSS v. 29 after appropriate data cleaning. Statistical significance for all analyses was set at *p* < 0.05.

The inclusion of Aboriginal and Torres Strait health practitioners started at conceptualisation and continued to data analysis, result interpretation, and manuscript writing. This ensures that the whole process of research is culturally appropriate for Indigenous peoples. The study received ethics approval from the University of Southern Queensland Human Research Ethics Committee (#22009821).

## 3. Results

The results are presented with different groups of findings.

### 3.1. Age and Sex

During the period of 1 July 2020 to 30 June 2021, there were 65,347 recorded cases presenting to the PUCC (missing *n* = 73 in age or sex or Indigenous status) with an average age of 38.45 years old (median 36 years and range 0 to 106). Twenty five percent of cases were under 21 years old and seventy-five percent under 55 years old. There were 3858 Indigenous patients with a median age of 28 years old and a range of 1 to 95 years old, and 35,361 non-Indigenous patients with a median age of 37 years old and a range of 0 to 106 years old (*p* < 0.001 compared with the Indigenous group) (Table 1).

The proportion of Indigenous patients attending the PUCC was 9.8% (*n* = 3858), with non-Indigenous patients at 90.2% (*n* = 35,361) of 65,429 completed cases (missing 26,201). The number of Indigenous females and males who attended PUCC was 2245 (58.4%) and 1602 (41.6%) (Table 1).

### 3.2. Residential Areas

Among the completed cases of 64,237 (missing 1183) who attended the PUCC, 25% lived in suburbs with an IRSAD score less than 898.00, and 75% in those with less than 941.00. Indigenous patients (*n* = 3769) lived in lower IRSAD suburbs (Median = 898.00; Mean Rank = 17,984.31) compared to non-Indigenous (Median = 929.00; *n* = 34,860; Mean Rank = 19,458.87; *p* < 0.001), where the lower the score, the higher the disadvantage (Table 1).

### 3.3. Patient Types

The existing patients are the individuals who visited the GP service before the establishment of the PUCC, and the new patients never visited the GP service prior to the PUCC.

Among the completed cases of 65,420 (Table 1) patients, 56,299 (86.1%) were existing patients and 9121 (13.9%) were new patients. Among the new patients, there were 525 (13.6%) Indigenous and 3397 (9.6%) non-Indigenous patients (*p* < 0.001).

### 3.4. Time of Arrival

After-hours care was defined based on the Royal Australian College of General Practitioners definition as Monday to Friday (6 p.m.–07:30 p.m.), Saturdays after 12 p.m., Sundays, and public holidays. Other times are in-hours or business hours.

Peak hours of attendance (*n* = 39,219) were from 11 a.m. to 12 p.m. and 2 p.m. to 3 p.m. for Indigenous patients, and 10 a.m. to 12 p.m. for non-Indigenous patients (*p* < 0.001; Table 2). Among the Indigenous patients, 69.1% attended the PUCC at business hours (30.9% at after-hours), while it is 70.1% for non-Indigenous patients (29.9% at after-hours; *p* > 0.05).

Mondays had the highest proportion of presentations for both groups (*n* = 10,861, 16.6%, *p* < 0.001) with Sundays (*n* = 7014, 10.7%) being the lowest (Table 3), and there was no significant difference between the two groups.

### 3.5. Reasons for Attendance

The top clinical reason for attending PUCC was AECC category E20 Injuries and other externally caused morbidity (*n* = 5154, 7.9%) for Indigenous (*n* = 305, 27.2%) and non-Indigenous (2555, 24.0%) patients from the 19,382 completed cases (*n* = 46,038 missing, 70.4%; *p* = 0.01; Table 4). These case presentations were classified into superficial injuries, such as wounds, abrasions, and/or minor fractures to the ankle and foot, wrist and hand, lower and upper arm, trunk or body region and the eyes, face, and head.

The second most frequent clinical reason for attending PUCC for Indigenous patients was AECC E06 Digestive issues (*n* = 113, 10.1%; *p* = 0.01; Table 3) consisting of nausea and vomiting, changes in bowel habits (i.e., diarrhoea or constipation), episodes of gastroesophageal reflux, acute gastroenteritis, and acute abdominal pain. This differed from non-Indigenous patients who presented with AECC category E08 Musculoskeletal issues (*n* = 1237, 11.6%; *p* = 0.01; Table 3) consisting of strain, pain or dislocation of ligaments or joints of the upper arm and shoulder, internal derangement of knee, inflammatory polyarthropathies, and dorsalgia.

## 4. Discussion

To the best of our knowledge, this is the first study to investigate Indigenous people’s use of a primary urgent care centre in a co-located GP service in a regional place in Australia. Our findings showed that the PUCC patients (both Indigenous and non-Indigenous) in 2020–2021 were predominately young adults, mostly from socioeconomically disadvantaged communities, largely GP service patients prior to the establishment of PUCC, and the leading reason for attending PUCC was superficial injuries. Compared with non-Indigenous patients, Indigenous patients were younger (28 vs. 37 years old), from more socioeconomically disadvantaged communities (IRSAD score 898 vs. 929); they represented a higher proportion of new patients (13.6% vs. 9.6%), and the second leading reason to attend PUCC is digestive issues while it is musculoskeletal issues for non-Indigenous patients. Times of arrival on a days or days in a week were similar between Indigenous and non-Indigenous patients, with Monday the busiest day of the week and 11 am–12 noon the busiest time of the day.

Among the PUCC users, approximately 10% were Indigenous patients while the proportion of Indigenous people in the population was 3.8% [23,24] and their use of the co-located GP service was only 3.5% (internal information). Our findings in a primary urgent care setting agree with a previous study in urgent care at hospital emergency departments (EDs) in Canada where the proportion of Indigenous patients was 9.4% while their proportion in the population was 4% [8], and the rate is higher outside urban or regional areas [12]. This confirmed the additional need for Indigenous people to access urgent healthcare.

This relatively higher rate of PUCC or hospital ED use by the Indigenous people could be due to a wide range of reasons, such as the difference in fee for the service, where PUCC or ED service was free for all patients while GP service was not, especially considering in this study that Indigenous people were from more disadvantaged suburbs. It needs to be noted that 3.5% access to GP services among Indigenous people does not mean that they had timely and quality care in primary healthcare settings. This is mainly because their health need is more urgent than that of the general population, and they are one of the most vulnerable and underserved groups in Australia [1,2,3,10,25]. Research findings actually support the lack of Indigenous access to timely primary healthcare services which consequently increased their need to access urgent care [8], at least partly due to the systemic drivers such as racism and lack of cultural safety in mainstream primary healthcare settings.

Our finding of more females than males among the Indigenous PUCC patients (approximately 60% vs. 40%) aligns with previous studies in urgent care at hospital EDs and primary healthcare settings [8,12,26]. McLane et al. [8] reported similar findings and co-interpreted their results with Indigenous Elders and Indigenous Health Directors to suggest that this may be due to Indigenous men having greater reluctance to seek care. Future research is needed to explore the reasons behind this “reluctance” so that future policy and practice could be developed and implemented to improve the healthcare access for Indigenous men.

Regarding the reasons for attending PUCC, the number one is injury for both Indigenous and non-Indigenous patients, and number two is musculoskeletal issues for non-Indigenous patients and digestive issues (10% of the Indigenous patients) for Indigenous patients. Our findings support a previous study of urgent care in Australian hospital EDs where the number one and two reasons for attending EDs for Indigenous patients are also injury and digestive issues [12].

Digestive issues include nausea and vomiting, changes in bowel habits (i.e., diarrhoea or constipation), episodes of gastroesophageal reflux, acute gastroenteritis, and acute abdominal pain. The comprehensive reasons underlining this high incidence of acute digestive health problems among Indigenous people in Australia are still unknown, although we could anticipate the potential environmental, cultural, and personal behavioural reasons. Future research is urgently needed to explore the whole contributing factors so that the health industry would work with the food industry to improve Indigenous digestive health collectively.

It is important to highlight that a PUCC provides urgent care for lower acuity injury and diseases, equivalent to Australian Triage Scale (ATS) 3 to 5 [27], with life-threatening presentations being diverted to hospital EDs. The Australian government report demonstrated that, from April 2023 to June 2023, this PUCC treated over 5000 patients at lower acuity (70% of ATS 5 and 30% of ATS 4), which may have attended the hospital EDs potentially [13]. This is significant for policy makers who aim to improve population’s timely access to urgent care in their communities with GP services, rather than hospital EDs, especially for underserved and vulnerable populations, such as Indigenous peoples. As this model of care is rolling out in Australia, it is vital for future research to comprehensively evaluate its effectiveness, efficiency, and equity, especially for those PUCCs which are co-located with community GP services, under the leadership and guidance of Indigenous communities.

Future studies could explore how we can improve Indigenous patients’ experiences at PUCCs, such as adopting the integration of culturally safe and supportive environments and service re-orientation at hospital EDs [5], building broader community relationships [28], and providing Indigenous traditional practitioner roles [28,29]. The PUCC model of care strategically addresses the location of care (free access for socieconomically disadvantaged areas within a GP service co-location), services of the facility (accreditation and trained staff to provide personalised care), and the goal of reducing hospital ED use [28,30]. It will be essential to comprehensively evaluate the PUCC model in a primary healthcare setting for its clinical safety and cost-saving, which is important for future policy makers, health managers, and clinicians. In addition, it would be interesting and important to address in future PUCC evaluation projects the characteristics of ambulance-transferred patients to hospitals, the patients who attended EDs after visiting PUCCs, whether disproportionate numbers attending PUCCs would be better seen in EDs, and whether disproportionate levels of digestive issues would be seen in local hospital EDs, as well as PUCCs’ effectiveness in reducing ED attendance.

### Limitations

This study is a secondary data analysis using medical record data. The accuracy and completeness of the data and missing data are significant limitations of our study due to its study design.

The high proportion of missing data in Reasons to Attend PUCC is partly due to the dataset records of “Administrative Activities” where they cannot be ICD-10-classified. Therefore, we need to interpret the findings with caution due to its limitation in representing the all patients attending the PUCC, although the study did have 19,382 cases for data analysis. The other potential reasons for the high proportion of missing data may include software limitations as Best Practice is primarily designed for clinical and administrative workflows, including Medicare claims management; human error in data entry as the software relies on clinicians to manually input the data; and the prioritisation of clinical care whereas data entry related to research may be secondary.

During our data collection period of the COVID-19 outbreak, patients exhibiting respiratory symptoms or illness (closely associated with COVID-19) were diverted to a separate quarantined respiratory clinic [29]. These specialised COVID-19 visits were not incorporated into the data analyses as COVID-19 patients were not part of the study.

It is important to note that the unique characteristics and location of the PUCC provide a distinctive environment to study the focus population; however, the results should be treated with caution regarding the appropriate generalisation to other PUCCs in Australia and other countries.

## 5. Conclusions

Indigenous patients accessed a primary urgent care centre at a higher rate for different health reasons compared to non-Indigenous patients in a regional Queensland primary healthcare setting. Co-located PUCCs within GP services could provide an opportunity to increase Indigenous patients’ access to acute care. Future research could comprehensively evaluate this PUCC model and Indigenous patients’ experiences to provide insights for policy makers and clinicians in Australia and other countries.

## Figures and Tables

**Table 1 ijerph-22-00998-t001:** PUCC use for Indigenous and non-Indigenous patients from 1 July 2020 to 30 June 2021 in North Moreton Bay, Australia.

	All Cases	Non-Indigenous (90.2%, *n* = 35,361)	Indigenous (9.8%, *n* = 3858)	Statistical Significance
**Age (years)** *n* = 65,347 (73 missing)	Median 36 (0 to 106) 25% < 21 years 75% < 55 years	Median 37 (0 to 106) Mean Rank 20,074.3 *n* = 35,361	Median 28 years (1 to 95) Mean Rank 15,353.95 *n* = 3858	Mann–Whitney U Test (U) U = 84,631,213 z = 24.59 *p* < 0.001
**Sex** *n* = 65,420 (980 missing)	34,426 female (53.4%) 30,014 male (46.6%)	19,147 female (54.2%) 16,170 male (45.8%)	2245 female (58.4%) 1602 male (41.6%)	Chi-Square (X^2^) 23.851 (1, *n* = 39,164) *p* < 0.001, phi = 0.025
**Relative Index of Socioeconomic Disadvantage** *n* = 64,237 (1183 missing)	Median 929.00 (713 to 1163) 25% < 898.00 75% < 941.00	Median 929.00 (713.00 to 1157) Mean Rank = 19,458.87 *n* = 34,860	Median 898.00 (800 to 1126) Mean Rank = 17,984.31 *n* = 3769	Mann–Whitney U Test U = 70,709,058.500, z = 8.144, *p* < 0.001
**Patient Type** Existing New *n* = 65,420 (0 missing)	56,299 (86.1%) 9121 (13.9%)	31,964 (90.4%) 3397 (9.6%)	3333 (86.4%) 525(13.6%)	Chi-Square (X^2^) 61.44 (1, *n* = 39,219) *p* < 0.001 phi = 0.040

**Table 2 ijerph-22-00998-t002:** Percentage in the time of day to attend PUCC for Indigenous and Non-indigenous patients from 1 July 2020 to 30 June 2021 in Moreton Bay X^2^ _(11, *n*=39,219)_ = 56.569; *p* < 0.001; phi = 0.038.

Time of Day	Indigenous Patients (%)	Non-Indigenous Patients (%)
0800–0859	4.6	6.5
0900–0959	7.0	9.2
1000–1059	9.2	10.6
1100–1159	10.4	10.4
1200–1259	7.3	6.7
1300–1359	9.8	9.5
1400–1459	10.8	9.5
1500–1559	9.8	9.9
1600–1659	7.6	7.8
1700–1759	7.3	7.2
1800–1859	8.6	6.9
1900–1959	6.6	5.9

**Table 3 ijerph-22-00998-t003:** The percentage to attend PUCC on each day of week for Indigenous and non-Indigenous patients from 1 July 2020 to 30 June 2021 in Moreton Bay X^2^ _(11, *n*=39,219_) = 4.90; *p* = 0.557; phi = 0.011).

	Indigenous Patients	Non-Indigenous Patients
Monday	16.4	16.8
Tuesday	16.0	16.4
Wednesday	15.9	16.3
Thursday	16.9	15.6
Friday	13.0	12.9
Saturday	11.4	11.8
Sunday	10.4	10.3

**Table 4 ijerph-22-00998-t004:** The percentage for the reasons to attend PUCC for Indigenous and non-Indigenous patients from 1 July 2020 to 30 June 2021 in Moreton Bay X^2^ _(15, *n*=11,758)_ = 30.518; *p* = 0.01; phi = 0.051.

	Indigenous Patients (%)	Non-Indigenous Patients (%)
E20 Injuries and other external morbidity	27.2	14.0
E60 Other factors influencing health status	12.3	13.2
E06 Digestive	10.1	8.6
E08 Musculoskeletal	9.5	11.6
E03 Ear, nose, mouth, and throat	9.1	9.7
E19 Mental and behavioural	7.8	6.1
E09 Skin subcutaneous and breast tissue	7.0	7.1
E05 Circulatory	3.6	4.6
E14 Obstetrics	2.8	2.4
E11 Kidney and urinary tract	2.7	3.6
E50 General symptoms without diagnosis	2.2	2.8
E01 Nervous system and neurological	1.9	2.0
E04 Respiratory	1.8	1.1
E10 Endocrine nutritional and metabolic	1.0	1.4
E02 Eye	0.6	1.2
E18 Infectious Diseases	0.4	0.6

## Data Availability

Our research data, deidentifiable patient data, is available when requested.

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
