# Peer review of "Indigenous People’s Use of a Primary Urgent Care Centre at a GP-Led Primary Healthcare Service in Regional Queensland in 2020–2021"

_ijerph, 2025, doi:10.3390/ijerph22070998_

Round 1

Reviewer 1 Report

Comments and Suggestions for Authors

Overview: This research article portrays the results of a cross-sectional study about the use of a primary urgent care center (PUCC) by Indigenous patients in regional Queensland (Australia). With a large sample from medical records, it describes the characteristics of these users, which is essential to comprehensively evaluate the PUCC model in a primary healthcare setting, and especially for underserved and vulnerable populations, such as Indigenous peoples.

Recommendation: Accept for publication with minor revision

Comments to Authors

Major comments:

  1. One of the main possible sources of bias is the fact that this study was only conducted in a single center, despite the fact that there seemed to be more than one funded center by this new program for new primary urgent care centers (PUCCs). Although the unique characteristics and location of said center provides a distinctive environment to study the focus population, the results should be treated with caution. And this limitation should be noted by the authors in the discussion section.

Minor comments:

  1. In the methods section, the type of study is not specified. Although one can deduce by the description that we are talking about a retrospective cross-sectional study, it should be mentioned at the beginning of the methodology section. This is also applicable to the title; being the type of study mentioned in at least one or the other.

  1. Line 110. The management and referral process is explained quite clearly, it could also be more complete if it included the information about what type of hospital were the patients transferred to, such as a tertiary hospital or other. The characteristics of said transferred patients would be an interesting addition.

  1. The methods section is clear. However, some details could be added to better describe the methodology used to gather the data, such as who was responsible or how the participants were selected (was it randomly or following some kind of pattern?) or were all patients included. It should also be addressed if there were cases in which the same patient returned to the ED (a common occurrence in primary care), and how the data for such cases was handled.

  1. This point follows on from #4 above, regarding expanding a bit about the statistical analysis as well. It should also be put together in the same paragraph all the information regarding the statistical analysis, such as which tests were carried out and which software was used.

  1. Although it is mentioned (line 130-131) that the study received ethics approval from a research ethics committee, it should be included that the patients signed an informed consent (line 349) for their personal data to be used in this study, regardless of the fact it could be obtained through their medical records and that it is anonymous.

  1. Line 137. I believe there is a typo after the date June 30th just as a helpful observation.

  1. Line 145. It mentions missing 26201, but it does not give any explanation whatsoever. It is mentioned that there were some missing information, but a third seems like a bit too much.

  1. Following the point #8 above, in Table 1, it gives the characteristics (age, sex…) of the overall population that attended the center, disregarding the missing 26201 and the fact that those numbers are not representative of the patients that were indeed categorized and analyzed (35361+3858) in the statistical analysis. Although it is interesting to know the characteristics of the population that attends and uses these centers, it does not mean it is representative of the sample studied. This data should also appear.

  1. Following the point #4 above, nowhere in the article are the potential sources of bias mentioned or addressed. Nor it is explained how the possible information bias was tackled (or at least mentioned as a limitation further down). It could be described if any efforts were made to address potential sources of bias.

  1. Regarding Table 2 and line 165, it would be interesting to explain in the text why the data was only recorded between 8am and 8pm. Were these the times when the center was open? What happened with the patients outside of this timeframe that was different?

  1. The interpretation of the results and the conclusions drawn from these results are satisfactory. Nonetheless, in my opinion it would be interesting to delve deeper into future recommendations, suggestions and studies that go beyond just description.

  1. I would suggest including the DOI of the different papers cited in this article in the references section, for clarity.

Personal Opinion: It is an interesting article since it taps into the least known and studied types of patients, in comparison to other populations. The results highlight the usage trends and how this population might be more at risk. If the few changes are implemented, its publication would add value to the existing literature on the topic, and aid to shape future programs and policies. These types of studies are important to identify rapid shifts of usage in a fast-changing world in the years to come, which is worrisome considering how at risk the Indigenous populations are and the lack of studies about their health outcomes.

Author Response

Reviewer 1: comments

Responses

Overview: This research article portrays the results of a cross-sectional study about the use of a primary urgent care center (PUCC) by Indigenous patients in regional Queensland (Australia). With a large sample from medical records, it describes the characteristics of these users, which is essential to comprehensively evaluate the PUCC model in a primary healthcare setting, and especially for underserved and vulnerable populations, such as Indigenous peoples.

Recommendation: Accept for publication with minor revision

We are grateful for the reviewer’s confirmation in the content of this manuscript.

Major comments:

One of the main possible sources of bias is the fact that this study was only conducted in a single center, despite the fact that there seemed to be more than one funded center by this new program for new primary urgent care centers (PUCCs). Although the unique characteristics and location of said center provides a distinctive environment to study the focus population, the results should be treated with caution. And this limitation should be noted by the authors in the discussion section.

Thank you for your insightful comment and this limitation has been added to the Discussion section, line   341-343.

Minor comments:

In the methods section, the type of study is not specified. Although one can deduce by the description that we are talking about a retrospective cross-sectional study, it should be mentioned at the beginning of the methodology section. This is also applicable to the title; being the type of study mentioned in at least one or the other.

Thank you for your eyes on details, this is greatly appreciated. We now added the study design at the beginning of the Methods section, line 104.

Line 110. The management and referral process is explained quite clearly, it could also be more complete if it included the information about what type of hospital were the patients transferred to, such as a tertiary hospital or other. The characteristics of said transferred patients would be an interesting addition.

Thank you for this important feedback. We added the hospital information (line 110) which would satisfy the readers’ needs.

We totally agree with you that the characteristics of said transferred patients would be an interesting addition. So, we added this important information to the section of future recommendations, line 320 -323, as this element is not the focus of this manuscript.

The methods section is clear. However, some details could be added to better describe the methodology used to gather the data, such as who was responsible or how the participants were selected (was it randomly or following some kind of pattern?) or were all patients included. It should also be addressed if there were cases in which the same patient returned to the ED (a common occurrence in primary care), and how the data for such cases was handled.

Thank you very much for this important feedback. More details have been added to the Method section, including that “all attending patients were included” line 113.

We agree with the Reviewer that it is a common occurrence in primary care where the same patient could return to ED after visiting the PUCC. This is such an important part in evaluating the effectiveness of PUCC in reducing hospital ED use. Despite that this is not part of the objectives in this manuscript, future comprehensive evaluation studies should not miss part at all. Therefore, this has been added to the future recommendation part, line 321.

This point follows on from #4 above, regarding expanding a bit about the statistical analysis as well. It should also be put together in the same paragraph all the information regarding the statistical analysis, such as which tests were carried out and which software was used.

Thank you very much for this comment.

This required information has been added to the Method section, line 127.

Although it is mentioned (line 130-131) that the study received ethics approval from a research ethics committee, it should be included that the patients signed an informed consent (line 349) for their personal data to be used in this study, regardless of the fact it could be obtained through their medical records and that it is anonymous.

We fully agree with the reviewer’s comment and thank you for picking up this oversight in the manuscript.

In the process of health practitioners entering the data, the patients do have a choice to tick “consent” or not for their information to be used for future health research.

When we extracted the data from the database, only the patients with the tick of “consent” were available for us to extract from.

This important information was added in line 113-114.

Line 137. I believe there is a typo after the date June 30th just as a helpful observation.

Thank you and this typo has been corrected.

Line 145. It mentions missing 26201, but it does not give any explanation whatsoever. It is mentioned that there were some missing information, but a third seems like a bit too much.

Thank you for your important comment.

We appreciate the reviewer’s feedback, and added some explanations in this section of limitation, including software limitation, human error in manually entering the data, missing clinical context, and prioritization of clinical care for clinicians.

These details are in line 332 -336.

Following the point #8 above, in Table 1, it gives the characteristics (age, sex…) of the overall population that attended the center, disregarding the missing 26201 and the fact that those numbers are not representative of the patients that were indeed categorized and analyzed (35361+3858) in the statistical analysis. Although it is interesting to know the characteristics of the population that attends and uses these centers, it does not mean it is representative of the sample studied. This data should also appear.

Thank you for the reviewer to point out this important and missing information in the manuscript.

We added this part to the section of limitations and ensure that readers understand the limitation of a high proportion of missing patient data and consequently the limitation in its representativeness for the whole patients in PUCC.

This added information is in line 330.

Reviewer 2 Report

Comments and Suggestions for Authors

Thank you for the opportunity to review your paper.  The topic was interesting and added to the existing body of work. Strengthening the integration of your findings into the existing body of knowledge is required to provide some reasons why and how the differences in usage occur.

Comments on the Quality of English Language

There are some editorial revisions required throughtout to improve clarity. Some have been noted in the table, but there were others

Author Response

Reviewer’s feedback

Responses

Line number

Comment

64 & 66

Extra space after full stop

The extra space has been removed. Thank you.

81

Discharge or discharged?

Thank you, it is “discharged”. This has been corrected.

247

How does indigenous usage compare with ED usage generally?

This is a good question, but to compare the use of PUCC with hospital ED may not be completely appropriate due to the risk of comparing “apples” and “oranges”.  

However, the current literature shows that in Australian hospital EDs, compared with non-Indigenous patients, Indigenous patients tend be triaged into less urgent categories (e.g., semi-urgent or non-urgent), and high rate of Leaving Without Being Seen or at Own Risk.

255

Ref 24 is on a Canadian population. Is this the correct reference?

Yes, it is correct, thank you for your question.

We are trying to show that shared scenarios in countries such as Australia and Canada.

256

Missing ‘s’ on finding?

Thank you for picking up this typo.

It is greatly appreciated and has been corrected.

258

Missing “the”?

This has been corrected. Thank you very much.

259

Clarity needed for the term outside urban. How does this differentiate from rural. It is interesting that there were similar rates between Indigenous peoples in Australia and Canada. What were the reasons

surmised by the Canadian Study, and how might that be reflected in an Australian context? How does this align with drivers of ED usage for Indigenous Australians i.e. Arabena K, Somerville E, Penny L,

Dashwood R, Bloxsome S, Warrior K, Pratt K, Lankin M, Kenny K, Rahman A. Traumatology talks–black wounds, white stitches.

Melbourne: Karabena Consulting, Australasian College for Emergency Medicine. 2020.

Thank you for this insightful question. The clarity has been added explaining the meaning of the agreed findings in different countries and different sectors of health services, in Line 268 -269.

261

How does this align with existing research on barriers to GP care for Indigenous Australians. It would be good to locate this important finding with previously established evidence on the barriers to GP care. Cost is important, but it is not the only driver of avoidable ED usage.

Thank you for this important question.

Barriers to GP care include a comprehensive list of items, and it is correct that cost is only of them. Other barriers include geographical locations, cultural differences, lack of cultural safety in healthcare system, racism, language, health literacy and communication issues.

It is correct that cost is not the only driver of avoidable ED usage for the Australian government to set up PUCCs, others include better access to urgent healthcare for wide communities especially for the vulnerable populations.

We agree that these are important questions and should be addressed in future studies.

266

Revise sentence for clarity. High health needs among an underserved population suggest they require greater access to primary care, but this is not the case. This paragraph could be an opportunity to explore the impact of systemic drivers of access such as racism, lack of cultural safety, cost etc. and how that increases ED demand. The numbers do seem to suggest that the presentations to the PUCC were appropriate – would some of the disproportionate numbers attending PUCC have been better seen in ED?

Thank you for this very important question.

We agree that the systemic drivers of access such as racism and lack of cultural safety in mainstream healthcare settings play important roles in leading to the high health needs among this underserved population. This has been added to line 279 -281.

Regarding whether some of the disproportionate numbers attending PUCC have been better seen in ED, it is such an important question, and we have added this part into the section of future recommendations, lines 328-329.

284

In the cited article were there any discussions on the reasons why these were the two primary reasons for attending ED?

Thank you for your question.

The sentence in this content is that “Our findings support a previous study of urgent care in Australian hospital EDs where number one and two reasons for attending EDs for Indigenous patients are also injury and digestive issues (12)” .

So, there is only one primary reason, which is number one, injury.

We hope this clarification helps.

285

Were the rates of digestive issues disproportionate to levels seen in the population, in ED’s, in the area? It would be of interest to note how this aligns with rates more generally or is this a unique finding for this

population and context.

Thank you for the question.

It is noted in the manuscript that digestive issues are the second reason for Indigenous people attending hospital Eds in Australia, but we did not have the ED data from the local hospital. This is an interesting question to explore, so we included this in the section of future recommendations in lines  328-329.

Broader comments

For an international audience it would be good to present more clearly the context to the PUCC urban/rural, population demographics, relative deprivation, general ED and Primary care usage.

It would be good to comment on whether the PUCC was effective at preventing unnecessary ED attendance for indigenous peoples.

Referencing was inconsistent in places e.g. line 296 ref inserted midpoint of the sentence rather than at the end when referring to the same document.

Thank you for the constructive comments.

We included the content that whether the PUCC was effective at preventing unnecessary ED attendance for indigenous peoples in the section of future recommendations as it is not within the scope of the current study.

The reference in line 296 has been moved to the end of the sentence.

Round 2

Reviewer 2 Report

Comments and Suggestions for Authors

 Thank you for the clear identification of the changes made in response to the feedback.  I look forward to seeing your article in print in the near future.